# Participation in a Voluntary Blood Donation Program as an Opportunity to Assess and Enhance Tetanus Immunity in Adult Blood Donors with an Outdated or Unknown Vaccination Status

**DOI:** 10.3390/vaccines13080884

**Published:** 2025-08-21

**Authors:** Katarzyna Tkaczyszyn, Małgorzata Szymczyk-Nużka, Leszek Szenborn

**Affiliations:** 1Clinical Department of Paediatrics and Infectious Diseases, Faculty of Medicine, Wroclaw Medical University, Chalubinskiego 2-2A Str., 50-368 Wroclaw, Poland; leszek.szenborn@umw.edu.pl; 2Regional Blood Transfusion Center, Czerwonego Krzyża 5/9 Str., 50-345 Wroclaw, Poland; malgorzata.szymczyknuzka@rckik.wroclaw.pl

**Keywords:** vaccination, tetanus, blood donor, post-exposure prophylaxis, immunity

## Abstract

**Background/Objectives:** Booster vaccination coverage in the adult population in Poland remains insufficient. The objective of this study was to utilize the opportunity of a visit to the Regional Blood Transfusion Center in Wroclaw—associated with blood donation—as a means to remind individuals about the need for tetanus booster vaccination and to assess tetanus immunity in healthy adults (30–40 years after their last mandatory dose) who had not received booster immunizations. **Materials and Methods**: A total of 97 blood donors aged 50 to 64 years (median age: 54 years) were enrolled, of whom 78% were male. 1. Tetanus immunity was assessed by a single measurement of serum anti-tetanus IgG antibody concentration. 2. A questionnaire was used to collect data relevant to tetanus immune status. 3. Individuals with insufficient protection received a booster dose of the tetanus vaccine, and the post-vaccination serologic response was evaluated. **Results:** 1. In the study group, 10.3% of participants had no protective immunity against *Clostridium tetani*, while 5.2% exhibited uncertain protection. An additional 32% demonstrated antibody levels conferring only short-term protection. Satisfactory protection—defined as immunity lasting at least 3 years—or long-term protection (at least 5 years) was identified in 52.5% of patients. Although 72% of donors reported receiving mandatory childhood immunizations, only 5% could provide medical documentation. In this subgroup, a significantly higher geometric mean antibody concentration was observed (0.69 vs. 0.52 IU/mL; *p* = 0.04), and significantly fewer participants required immediate post-exposure prophylaxis (1/39 vs. 14/54; *p* = 0.003). 2. Among the 46 individuals eligible for a booster dose, 17 (37%) returned for vaccination. Of these, 16 (94%) achieved antibody titers consistent with long-term protection following a single vaccine dose. **Conclusions**: Tetanus immunity among adults is heterogeneous and difficult to predict due to the frequent lack of vaccination records and unreliable self-reported histories. A history of injury and associated surgical wound care involving injection often serves as the only indication of prior vaccination. A single booster dose is highly effective in eliciting a robust immune response in individuals vaccinated during childhood but lacking recent boosters. Rising vaccine hesitancy toward both mandatory and recommended immunizations in Poland negatively influences adult decisions regarding tetanus vaccination. Participation in voluntary blood donation programs presents a valuable opportunity for immunization education, immune status screening, and the implementation of effective catch-up vaccination strategies.

## 1. Introduction

The implementation and dissemination of adult vaccination programs in Poland remain a significant public health challenge. Due to long-standing systemic neglect, the limited scope of recommendations within the National Immunization Program (NIP), and the financial burden of vaccine costs placed on patients, adult vaccination coverage in Poland remains critically low [1]. Until 2019, the only free vaccine available to adults was the decennial tetanus and diphtheria booster; however, no structured reminder systems or public promotion efforts were in place to encourage uptake. During the coronavirus disease 19 (COVID-19) pandemic, adult vaccination coverage against SARS-CoV-2 in Poland lagged behind that in Western European countries [2,3]. Between 2022 and 2025, major policy changes were introduced to expand access to free vaccinations (e.g., against influenza, COVID-19, and RSV) and to subsidized vaccines (e.g., pneumococcal, herpes zoster, and high-dose influenza). These initiatives were supported by public education campaigns and the development of adult immunization schedules accounting for individual risk factors [4,5,6]. In 2024, the NIP was further extended to provide free vaccinations against selected infectious diseases to specific groups of immunocompromised adults [7].

Despite these initiatives, adult vaccination rates in Poland remain suboptimal and call for additional systemic and organizational efforts. One potentially effective venue for promoting adult immunization could be Regional Blood Transfusion Centers, as blood donors often participate voluntarily and conscientiously in healthcare initiatives aimed at protecting others and may therefore be more health-conscious and receptive to preventive measures. According to 2023 data, there were 23 Regional Blood Transfusion Centers in Poland, supported by 135 local blood donation sites, with a total of 639,972 registered blood donors [8]. Participation in voluntary blood donation programs presents a unique opportunity to assess post-vaccination immunity, including tetanus immunity among adults who were vaccinated in childhood but have not received boosters in decades. It also provides an opportunity to inform donors about the need for booster vaccinations.

Tetanus vaccination in Poland is mandatory (and free of charge), following the national schedule: four primary doses administered at 2, 4, 6–7, and 16–18 months of age, followed by booster doses at 6, 14, and 19 years [7]. This schedule aligns with international recommendations [9]. In adulthood, tetanus prophylaxis typically takes the form of post-exposure prophylaxis (PEP) after injury, and decisions regarding vaccine administration depend on the individual’s immunization history and the timing of their last booster dose.

Although not mandatory, tetanus booster doses (containing tetanus toxoid) are recommended in Poland every 10 years [7]. Despite long-standing mandatory childhood immunization programs, Poland continues to report several tetanus cases and related deaths each year. According to the National Institute of Public Health (NIPH), between 2014 and 2023, 2 to 17 tetanus cases and 0 to 4 deaths were recorded annually. During the COVID-19 pandemic (2020–2022), a decline in reported tetanus cases was observed, likely due to reduced healthcare utilization and pandemic-related public health measures [10]. In 2023, 13 tetanus cases were reported in Poland [11], all among adults, with the highest incidence observed in individuals aged 70 years or older. These findings are consistent with those of reports from the European Centre for Disease Prevention and Control (ECDC), which documented 56 tetanus cases across 26 EU countries in 2022 (incidence: 0.02 per 100,000), including 7 deaths. Over 80% of the reported cases occurred in individuals aged ≥65 years [12]. The vast majority of tetanus cases occur in patients with unknown or uncertain vaccination histories. Importantly, no cases of neonatal tetanus have been reported in Poland since 1984 [13].

According to the NIP 2025 chapter “Tetanus vaccination in individuals exposed to injury (post-exposure prophylaxis),” in the event of a low-risk wound, unvaccinated individuals, those with incomplete vaccination, or those with uncertain histories should receive a tetanus or tetanus–diphtheria vaccine and continue with a three-dose primary vaccination schedule (administered at 0, 1, and 6 months). In high-risk wounds, a weight-adjusted dose of tetanus-specific immunoglobulin (250/500 IU) is additionally recommended [7]. As a result of these guidelines, many patients are—or should be—vaccinated annually using the complete three-dose schedule. Nonetheless, tetanus prophylaxis in Polish adults remains insufficient. Post-vaccination antibody levels are not routinely monitored, leaving it unclear whether individuals are adequately protected following a booster dose. In the late 1980s, Gałązka et al. studied tetanus immunity in patients aged 10–90 using the passive hemagglutination method [14]. Zakrzewska later examined anti-tetanus antibody levels in blood donors using modern enzyme immunoassays [15]; however, neither study assessed the immune response following the administration of a booster dose.

The primary objective of this study was to utilize a visit to a Regional Blood Transfusion Center (during blood donation) as an opportunity to remind individuals about the importance of tetanus booster vaccination. Secondary objectives included evaluating tetanus immunity among healthy adult blood donors (30–40 years after their last mandatory vaccination), who had not received booster doses after age 19 or had unknown or uncertain vaccination histories, and assessing the immune response after the administration of a single booster dose.

## 2. Materials and Methods

This study was offered to voluntary blood donors presenting for blood donation at the Regional Blood Transfusion Center in Wroclaw. Participation in this study was voluntary and free of charge. The primary inclusion criteria were as follows: age ≥ 50 years, consent to blood sample testing, and willingness to respond to questions regarding tetanus vaccination history and factors potentially influencing tetanus immunity. Blood donors aged 50–64 who presented at the blood donation center with the intention to donate blood received information about the study and an informed consent form during the pre-donation medical screening. If, after reviewing the study’s purpose, procedures, and conditions, a blood donor provided written informed consent and was subsequently deemed eligible to donate blood by a physician, a blood sample was collected for the assessment of post-vaccination antibody levels. A total of 97 participants aged 50 to 64 years (median age: 54 years) were enrolled. All blood donors who were invited to participate in the study agreed to take part in the project.

This study was conducted in two phases:

1. Assessment of tetanus immunity based on a one-time measurement of serum IgG-specific anti-tetanus antibody levels, performed at the research laboratory of the Department of Paediatrics and Infectious Diseases, Wroclaw Medical University. Antibody levels were assessed using the commercial Enzyme-Linked Immunosorbent Assay (ELISA) test Tetanus Toxoid IgG, catalog no. EI 2060-9601G (Euroimmun, Lübeck, Germany). This was carried out for all participants. Additionally, each participant completed a custom-designed questionnaire consisting of 16 items related to variables potentially affecting tetanus immunity. A sample questionnaire is attached to the manuscript as Appendix A.

2. Implementation of prophylactic vaccination in the subgroup of donors found to have insufficient protection against *Clostridium tetani* infection (Table 1). All study participants whose antibody levels were considered insufficient were notified—either by letter or telephone—and were invited to the Department of Paediatrics and Infectious Diseases for a medical consultation and to be offered a booster dose of the tetanus vaccine, followed by a reassessment of the immune response. The follow-up antibody testing was conducted 4 to 6 weeks after vaccine administration.

Post-vaccination immunity was assessed using the commercial ELISA test Tetanus Toxoid IgG, catalog no. EI 2060-9601G (Euroimmun, Germany). The concentration of tetanus-specific IgG antibodies was measured and interpreted according to the international standard, with antibody titers expressed in international units (IU/mL).

This study was conducted in accordance with the Declaration of Helsinki, and the protocol was approved by the Bioethics Committee of Wroclaw Medical University, Wroclaw, Poland [Approval Code: 385/2015; Approval Date: 3 September 2015].

The obtained data, antibody concentrations (>0.1 IU/mL), and geometric mean concentrations in the groups and subgroups were analyzed using statistics. The significance of observed differences was assessed using the Mann–Whitney U test and the Chi-square (χ^2^) test. Statistical analyses were performed using STATISTICA 13.3 data analysis software (TIBCO Software, Inc., Palo Alto, CA, USA).

## 3. Results

### 3.1. Characteristics of the Study Population and Assessment of Tetanus Immunity Among Adult Blood Donors

In the study group, the majority of participants were male (78%). The largest subgroup comprised participants who had completed secondary education (59%), with the following subgroups comprising those with higher education (29%) and primary education (7%). Most participants were employed at the time of the study (84%) and resided in urban areas (82%).

Of 97 individuals, 10 (10.3%) had no protective immunity against tetanus, while 5 (5.2%) exhibited uncertain protection. Short-term protective immunity was found in 31 individuals (32%). Good protection—defined as lasting at least 3 more years from the time of antibody testing—was identified in 32 participants (33%). Long-term protection (>5 years) was confirmed in 19 participants (19.5%)—see Figure 1.

The majority of participants (72%) reported having received mandatory tetanus vaccinations as part of the National Immunization Program (NIP); however, only 5% stated that they possessed documentation of their immunization history. Based on the information collected, 18% of donors may have received a tetanus booster dose, while as many as 40% likely received a booster vaccination during surgical wound management. None of the participants had undergone a repeated primary vaccination series.

Participants were asked about activities potentially associated with an increased risk of *Clostridium tetani* exposure. Over half (55%) reported working in a garden or allotment, while 45% admitted to having sustained a wound contaminated with soil in the past.

No statistically significant differences in tetanus immunity status were observed based on sex, urban versus rural residence, level of education, or self-reported prior prophylactic tetanus vaccination. Detailed results are presented in Table 2.

A statistically significant association was found between tetanus immunity status and a likely history of unintentional post-exposure prophylaxis. This subgroup was identified based on a positive response to the following question: “Have you ever received an injection (vaccine) following an injury or accident?” Within this group, a significantly higher geometric mean antibody concentration was observed (0.69 vs. 0.52 IU/mL; *p* = 0.04), as well as a significantly lower number of donors requiring immediate post-exposure prophylaxis (1/39 vs. 14/54; *p* = 0.003).

### 3.2. Effectiveness of a Single Tetanus Booster Dose Administered Many Years After the Last Vaccination

Blood donors who were found to have no tetanus immunity or uncertain post-vaccination protection were classified as unprotected and eligible for a free booster dose. Of the 46 individuals qualified to receive the tetanus booster (with individually scheduled appointments arranged by phone), 17 (37%) responded to the invitation and were vaccinated. Despite receiving information about the benefits and safety of booster vaccination, as well as multiple reminders, the remaining invitees declined vaccination.

Among the 17 vaccinated individuals, 16 demonstrated a significant increase in specific anti-tetanus antibody levels consistent with long-term protection after a single booster dose containing tetanus toxoid. In one case, no immune response was observed following the booster dose; this was a healthy 64-year-old woman.

## 4. Discussion

In the studied group of blood donors aged ≥50 years, a high proportion of individuals demonstrated protective levels of anti-tetanus antibodies—89.7%. Insufficient protection, defined as antibody titers < 0.1 IU/mL, was observed in 10.3% of participants. These results are consistent with findings from studies conducted by other authors among blood donor populations. Eslamifar et al. reported a high prevalence of tetanus immunity in Iranian blood donors—96% in the 18–71 age group, including 92% among those over 50 years of age [16]. Similarly, Martin et al. found protective immunity in 96.3% of donors (98.7% of men and 94.1% of women aged 18–64 years, with 99% of men and 85% of women ≥ 50 years) [17]. Yuan et al. reported that 82.5% of blood donors aged 20–70 years had protective immunity [18]. All of these studies used comparable criteria for assessing post-vaccination immunity, applying enzyme immunoassays and the same antibody titer thresholds. It is important to note the age range of the participants in our study. The lower inclusion limit was 50 years—approximately 30 years after the last mandatory tetanus vaccine dose—while the upper limit corresponded to the maximum eligible age for blood donation in Poland (65 years). This focused age range is a strength of our study, as it highlights a high rate of preserved immunity in older adults. In contrast, the aforementioned studies [16,17,18] also included younger participants, which may have influenced the overall results. In Poland, tetanus immunity among blood donors was previously assessed in 1997, and, in that study, all participants (aged 20–59 years) had protective levels of anti-tetanus antibodies [15].

Studies conducted in the general population (not limited to blood donors) have shown more varied levels of tetanus immunity. Karabay et al. reported protective antibody levels in only 15.4% of nursing home residents, with a mean age of 71 years [19]. A Danish study found protective titers in 49% of participants aged 30–70 years [20]. In a study by Mizuno et al., conducted among Japanese travelers, 76% had protective antibody levels [21]. In an Iranian study involving trauma unit patients, 87.3% of individuals (mean age: 40 years) were immune [22]. Janout et al. reported protective titers in as many as 90.9% of participants aged over 60 years in the Czech Republic [23]. In a large U.S. study, the proportion of individuals with protective immunity ranged from over 80% in children aged 6–11 years to only 27.8% in those aged 70 or older [24]. Table 3 summarizes the results of studies evaluating tetanus immunity across different populations and countries.

As shown above, the proportion of individuals with protective immunity against tetanus varies across numerous studies; however, many authors report a common trend—protective anti-tetanus antibody levels tend to decline with age [19,22,24,25,26]. This highlights the importance of administering periodic booster doses of tetanus-containing vaccines. The Polish National Immunization Program (NIP), in line with global recommendations [9], recommends a booster dose every 10 years for individuals who have completed the primary immunization series or as post-exposure prophylaxis following an injury [7]. However, routine post-booster serological testing is not recommended. Furthermore, in cases of unknown or uncertain vaccination history, the NIP recommends repeating the full primary vaccination series. There is limited literature assessing the immune response to a tetanus booster, particularly in adults. In one such study, Mizuno et al. demonstrated booster effectiveness ranging from 61.5% to 100% in age groups of 20–39 years and 40–>50 years, respectively [21]. In our study, 17 out of 46 blood donors (37%) accepted the invitation and received a booster dose. This represents a relatively good outcome, especially when compared, e.g., to the percentage of individuals aged 65 or older who received a free influenza vaccine during the 2020/2021 pandemic season—only 22.94%. In the most recent 2024/2025 season, the uptake dropped to just 17.24% of those eligible [27]. These findings support the idea that blood transfusion centers are an effective setting for promoting adult vaccination awareness in Poland. Notably, 16 out of the 17 vaccinated individuals (94%) achieved protective levels of post-vaccination anti-tetanus antibodies.

Olander et al. conducted an analysis of anti-tetanus antibody levels among Finnish citizens and found that over 70% of individuals aged >50 years had protective titers. Notably, 76% of participants reached antibody levels >1 IU/mL, indicating long-term protection. Based on these findings, the authors proposed reconsidering current recommendations by extending the interval between booster doses from 10 to 20 years [28]. In our study, 19.5% of blood donors demonstrated long-term post-vaccination immunity despite having uncertain vaccination histories. Given the confirmed high effectiveness of a single booster dose, it is worth re-evaluating whether repeating the entire primary vaccination series is truly necessary—particularly in the context of post-exposure prophylaxis among patients with no documentation or an uncertain immunization history. As emphasized by the present study and others [18,21], accurately determining a patient’s vaccination status remains a major challenge. Adult patients rarely possess records of prior immunizations, and self-reported histories are often unreliable. In our study population, only 5% of blood donors reported having documentation of previous vaccinations. In this context, and in the era of digital healthcare, the implementation of an electronic vaccination record, accessible via mobile devices and integrated into national health platforms (such as Poland’s Internet Patient Account), would be of great practical value.

Another important issue is the frequent lack of awareness among patients regarding the receipt of tetanus prophylaxis—such situations most commonly occur in emergency department settings, for example, following accidents or significant wounds. In our study, 40% of blood donors reported having received an injection after an injury; however, many of them were uncertain about what the injection contained and lacked corresponding medical documentation. As such information is unreliable, it must be assumed that the presence of a so-called “trauma history” may indicate the possibility that the patient received a tetanus booster dose.

Given the growing number of individuals refusing vaccination in Poland—from 5340 cases in 2012 to 87,098 in 2023 [29]—and the associated rise in the incidence of vaccine-preventable diseases, accurate and consistent education regarding active immunization is becoming increasingly important. Adults presenting to Regional Blood Transfusion Centers always come into contact with a physician and are exposed to health-related information, including on bloodborne infections and healthy lifestyle practices. These encounters also provide an excellent opportunity to offer reliable information on available vaccinations. Thus, voluntary blood donation programs appear to be a valuable setting for patient education while simultaneously offering a chance to propose necessary immunizations—particularly against tetanus and influenza—that contribute directly to health protection.

## 5. Limitations of the Study

Our study has certain limitations and constraints. First and foremost, the study population was relatively small, consisting of only 97 participants. This limited sample size was primarily due to the strict inclusion criteria. In addition to the age requirement (50–64 years), a key criterion was the absence of tetanus vaccination within the past 30 years. During the recruitment period, 97 individuals who met these criteria and provided informed consent were enrolled. We acknowledge that a larger sample size would have enhanced the statistical power of our analysis and allowed for a broader generalization of the findings. Nevertheless, we consider this study a potential starting point for further research.

Another limitation may stem from the specific nature of the study population. Blood donors represent a self-selected group who are typically aware of the strict health requirements necessary for donation eligibility. As such, this population is routinely screened for various health conditions, including infectious diseases. However, when designing the study, we assumed that this heightened health awareness does not necessarily correlate with high vaccination coverage. Post-vaccination immunity against tetanus—as well as against other vaccine-preventable diseases—is not routinely assessed. In clinical practice, only anti-HBs antibody levels are commonly measured following hepatitis B vaccination. Therefore, we were interested in determining whether individuals who demonstrate proactive health behaviors through regular blood donation also exhibit a strong commitment to preventive care.

Despite the limitations mentioned above, we believe that our study provides valuable data on tetanus post-vaccination immunity among adults in Poland. To date, no prospective study has assessed whether—and how—the blood donation process could serve as an opportunity to evaluate and potentially improve post-vaccination immunity.

## 6. Conclusions

Tetanus immunity among adults is heterogeneous and unpredictable, primarily due to the widespread lack of vaccination records and the unreliability of self-reported histories. Injuries involving soil contamination are common and often underestimated in adults with an unknown vaccination status. In previously vaccinated individuals, such injuries should always prompt post-exposure prophylaxis. A helpful indicator of possible tetanus immunity is a history of injury requiring surgical management with the administration of an injection. A single booster dose is highly effective in inducing immunity in individuals who have not been vaccinated in many years. The current Polish NIP recommendation to repeat the entire primary tetanus vaccination series in individuals who received their last dose more than 10 years ago appears unjustified and unnecessary. In the era of digital health records, it is essential to develop an electronic vaccination history for every patient, which would provide access to reliable immunization data. Participation in voluntary blood donation programs offers a valuable opportunity for education on the need for booster vaccinations. The willingness of donors to participate in the study supports this approach. Blood donation centers could serve as an additional platform to deliver catch-up vaccinations in the adult population. The increasing prevalence of vaccine hesitancy in Poland—affecting both mandatory and recommended immunization programs—may have a detrimental impact on adult compliance with tetanus prophylaxis. Therefore, it is crucial to seize every clinical opportunity to educate patients and promote immunization.

## Figures and Tables

**Figure 1 vaccines-13-00884-f001:**
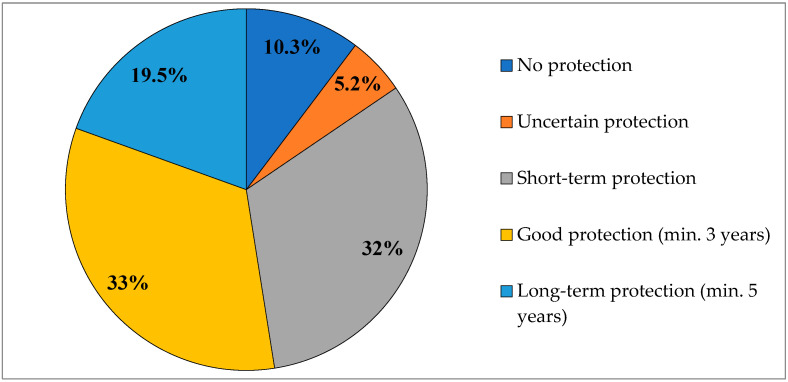
Tetanus immunity status in the study population.

**Table 1 vaccines-13-00884-t001:** Assessment of tetanus immunity status according to international criteria and interpretation (test used: Tetanus Toxoid IgG, Cat. No. EI 2060-9601G, Euroimmun).

Anti-Tetanus IgG Antibody Level [IU/mL]	Manufacturer’s Interpretation	Post-Vaccination Immunity Status
<0.01	No protection; primary immunization required	Non-protective
0.01–0.1	Uncertain post-vaccination protection; next dose recommended in 4–8 weeks	Non-protective
>0.1–0.5	Short-term protective immunity present; booster vaccination recommended	Protective
>0.5–1.0	Protective immunity present; next booster recommended in 3 years	Protective
1.0–5.0	Long-term protective immunity present; next booster in 5 years	Protective
>5.0	Long-term protective immunity present; next booster in 8 years	Protective

**Table 2 vaccines-13-00884-t002:** Tetanus immunity and demographic factors of the study population.

Variable		Immune N (%)	Non-Immune N (%)	Chi2 *p*-Value
**Sex**	Female	16 (76.2)	5 (23.8)	NS
	Male	66 (86.8)	10 (13.2)	
Place of residence	Urban	67 (83.7)	13 (16.2)	NS
	Rural	13 (86.7)	2 (13.3)	
Education level	Primary + secondary	57 (87.7)	8 (12.3)	NS
	Higher	22 (75.9)	7 (24.1)	
Additional vaccination	Yes	16 (94.1)	1 (5.9)	NS
	No	60 (81.1)	14 (18.9)	

Abbreviations: NS—not significant.

**Table 3 vaccines-13-00884-t003:** Summary of studies assessing tetanus immunity in various populations and countries.

Authors	Country	Population	Age Range (Years)	Proportion Immune (%)
Zakrzewska, 1997 [15]	Poland	Blood donors	20–59	100
Eslamifar et al., 2014 [16]	Iran	Blood donors	18–71	96 (92 in >50 age group)
Martin et al., 2005 [17]	Germany	Blood donors	18–64	96.3 (92 in ≥50 age group)
Yuan et al., 1997 [18]	Canada	Blood donors	20–70	82.5
Karabay et al., 2005 [19]	Turkey	General (nursing home residents)	60–≥76	15.4
Kjeldsen et al., 1988 [20]	Denmark	General	30–70	49
Mizuno et al., 2014 [21]	Japan	General	20–≥50	76
Afzali et al., 2015 [22]	Iran	General (trauma unit patients)	40.9 ± 3.7	87.3
Janout et al., 2005 [23]	Czech Republic	General	60–≥90	90.9
Gergen et al., 1995 [24]	United States	General	≥6	69.7 (~50 in ≥50 age group)
Liu et al., 2021 [25]	China	General	1 day—89	54.4 (<10 in >40 age group)
Bampoe et al., 2024 [26]	United States	General	≥6	93.8 (>90 in 50–69 age group)

## Data Availability

Study database is available on request at the Clinical Department of Paediatrics and Infectious Diseases, Wroclaw Medical University.

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
