# Peer review of "Participation in a Voluntary Blood Donation Program as an Opportunity to Assess and Enhance Tetanus Immunity in Adult Blood Donors with an Outdated or Unknown Vaccination Status"

_vaccines, 2025, doi:10.3390/vaccines13080884_

Round 1
Reviewer 1 Report
Comments and Suggestions for Authors
The study focuses on donors aged 50–64 years, yet there's limited elaboration on why this specific age band was chosen. While it aligns with the potential lapse since the last childhood booster, a deeper rationale perhaps linked to epidemiological trends or immune senescence would contextualize this choice. Consider citing age-specific tetanus incidence or immune decline data from Poland or Europe.
1) Only 17 of the 46 eligible participants (37%) returned for booster vaccination. The manuscript notes reminders were given, but it lacks information on:
-
How many reminders were sent?
-
Time between reminders?
-
Reasons for non-compliance?
Including qualitative data or a brief survey of non-responders could offer valuable insights into vaccine hesitancy and logistical barriers
2)How consent was obtained and documented
3)The distinction between "short-term," "good," and "long-term" protection in Figure 1 and Table 1 is informative, but potentially confusing.
4) The conclusion criticizes the Polish National Immunization Program’s recommendation to restart the primary series for adults with unknown vaccination histories. This is a significant claim. Strengthen this section by proposing alternative strategies ??
Author Response
Comment 1. The study focuses on donors aged 50–64 years, yet there's limited elaboration on why this specific age band was chosen. While it aligns with the potential lapse since the last childhood booster, a deeper rationale perhaps linked to epidemiological trends or immune senescence would contextualize this choice. Consider citing age-specific tetanus incidence or immune decline data from Poland or Europe.
Response 1. We appreciate the Reviewer’s comment regarding the age limits applied in our study. In Poland, the last dose of the primary tetanus vaccination is administered at the age of 19, with subsequent booster doses recommended every 10 years. In recent years, tetanus cases in Poland have been reported in patients aged approximately 50–60 years, i.e., at least 30 years after the last mandatory dose of vaccination; therefore, we adopted 50 years as the lower age limit. The upper age limit was determined by the maximum age of a blood donor eligible to present at the Regional Blood Transfusion Center (RCKiK) — 60 years for first-time donors and 65 years for repeat donors. Data on tetanus incidence in other European countries are provided later in the Introduction.
Comment 2. Only 17 of the 46 eligible participants (37%) returned for booster vaccination. The manuscript notes reminders were given, but it lacks information on: How many reminders were sent? Time between reminders? Reasons for non-compliance? Including qualitative data or a brief survey of non-responders could offer valuable insights into vaccine hesitancy and logistical barriers.
Response 2. We thank you for your valuable comments. All study participants whose antibody levels were deemed insufficient were informed either by letter or by telephone and were asked to report to the Department of Paediatrics and Infectious Diseases (the study site) for a consultation with a physician and to be offered a booster dose of the vaccine. Multiple attempts to contact them were made at short (several-week) intervals. The reasons for non-attendance remain unknown. We agree with the reviewer that supplementing the study with qualitative data or a brief survey among non-respondents could provide valuable insights into vaccine hesitancy or logistical barriers; this is an excellent suggestion to be implemented in future research.
Comment 3. How consent was obtained and documented?
Response 3. The informed consent form, together with the study information sheet, was presented to the study participant during the eligibility assessment for blood donation. Written consent was obtained by the physician responsible for qualifying the blood donor for donation.
Comment 4. The distinction between "short-term," "good," and "long-term" protection in Figure 1 and Table 1 is informative, but potentially confusing.
Response 4. We appreciate this insightful comment. To determine the immune status, we adhered to the terminology provided in the official interpretation guidelines of the assay (Tetanus Toxoid IgG, catalog no. EI 2060-9601G, Euroimmun, Germany). This approach was adopted to ensure methodological consistency and accuracy, as these exact terms were used in the interpretation of the test results assessing post-vaccination antibody levels.
Comment 5. The conclusion criticizes the Polish National Immunization Program’s recommendation to restart the primary series for adults with unknown vaccination histories. This is a significant claim. Strengthen this section by proposing alternative strategies ??
Response 5. We thank you for this comment. Our critique of the Polish National Immunization Program recommendations on repeating the full vaccination series in adults with unknown vaccination history is based on two main considerations. First, the absence of a central vaccination registry means that patients often lack documentation and cannot recall which vaccines they received or when. While the Internet Patient Account in Poland now stores certain medical records, vaccination data are not yet fully integrated. We hope that ongoing work in this area will soon allow physicians to verify immunization history, avoiding unnecessary doses and identifying those in need of a booster. Second, our study found that over 90% of participants (16/17) achieved long-term immunity after a single booster dose. We therefore suggest that in patients likely vaccinated against tetanus more than 10 years earlier, antibody testing after one dose could guide whether to complete the full series or limit vaccination to that single dose. We did not address in this study the optimal timing, setting, or cost-effectiveness of such testing.
Reviewer 2 Report
Comments and Suggestions for Authors
Dear authors,
Thank you for conducting this study and for giving me the opportunity to read this work.
Title of the Article: "Participation in a voluntary blood donation program as an opportunity to assess and enhance tetanus immunity in adult blood donors with outdated or unknown vaccination status."
Journal: Vaccines (ISSN 2076-393X)
Reference of the Ms: "vaccines-3769346"
The manuscript is interesting because according to the authors, there is a low tetanus protection in Poland, and that blood donation visits for education and screening is effective for identifying individuals who are at risk. Consequently, blood donation programs can be a good idea to help to improve tetanus vaccination rates. Thus, all this makes this article original and worthy of publication. Nevertheless, I think that some small adjustments are necessary.
Here are some comments:
- Line 6-7, please try to give more details regarding your institution, such as the country, or any other additional information.
- Line 8, given that you've chosen a structured abstract, I think you have to add "background" before the first sentence, and "objective" before the 2nd one, or "Background/objectives" in the Beginning.
- Line 9, please consider changing the term "inadequate" as it is not representative, maybe you mean just "low" or "insufficient" or "suboptimal"...etc.
- Line 37, please add at least 1 or 2 additional keywords to give the study a broader audience.
- Line 39, please write introduction in Bold
- Lines 40-44, before starting to write about Poland, I think it is better to put a more general background such as writing about vaccination in general or tetanus or both of them, instead of starting to write directly about the problematic in Poland.
- Line 47, please write "Coronavirus disease 19 (COVID-19)" instead of the directly the acronym, since it was the first time that this appeared in the text.
- Lines 115-119, there are results, please consider moving this part to the results section or to remove it to avoid repetition.
- Line 129, I would have appreciated the content of the 17 items of the questionnaire, maybe it would be interesting for the readers if it will be shared in Supplementary Material as an example.
- Line 140-142, please avoid copying exactly the same sentence as lines 312-314 (Institutional Review Board Statement).
- Please consider improving the quality of Figure 1 (Capitalize the first letters, avoid the overlap...etc.) . In addition, captions of figures must be put under the figure.
- In the whole manuscript, please consider following the instructions to authors in the journal website to avoid minor mistakes related to the layout.
- Line 196, same as my comment number 5
- The discussion section was well written, I would like to congratulate the authors, the structure is logical and fluid. Moreover, the limitations and perspectives are described indirectly within the flow, which will be pleasant for readers.
- Line 289, same as my comments number 5 and number 13.
- Lines 305-307, please try to rephrase this part as it doesn't read well.
- Regarding references, if possible, please consider adding more recent references, specifically from 2025.
Again, thank you for this work and good luck.
Author Response
Comment 1. Line 6-7, please try to give more details regarding your institution, such as the country, or any other additional information.
Response 1. We thank you for pointing this out. The affiliations have been revised accordingly, and the corrected version is presented below.
- Wroclaw Medical University, Faculty of Medicine, Clinical Department of Paediatrics and Infectious Diseases, Chalubinskiego 2-2A Str., 50-368 Wroclaw, Poland.
- Regional Blood Transfusion Center, Czerwonego Krzyża 5/9 Str., 50-345 Wroclaw, Poland.
Comment 2. Line 8, given that you've chosen a structured abstract, I think you have to add "background" before the first sentence, and "objective" before the 2nd one, or "Background/objectives" in the Beginning.
Response 2. We appreciate your comment. The abstract has been revised in accordance with the provided recommendations.
Background/Objectives: Booster vaccination coverage in the adult population in Poland remains inadequate. The objective of this study was to utilize the opportunity of a visit to the Regional Blood Transfusion Center in WrocÅ‚aw – associated with blood donation – as a means to remind individuals about the need for tetanus booster vaccination and to assess tetanus immunity in healthy adults (30-40 years after their last mandatory dose) who had not received booster immunizations.
Comment 3. Line 9, please consider changing the term "inadequate" as it is not representative, maybe you mean just "low" or "insufficient" or "suboptimal"...etc.
Response 3. We agree with the reviewer that the term originally used was not sufficiently precise. It has therefore been replaced with a more appropriate one, as shown below.
Booster vaccination coverage in the adult population in Poland remains insufficient.
Comment 4. Line 37, please add at least 1 or 2 additional keywords to give the study a broader audience.
Response 4. The following keywords have been added: post-exposure prophylaxis, immunity.
Comment 5. Line 39, please write introduction in Bold
Response 5. The revisions are indicated in bold font.
Comment 6. Lines 40-44, before starting to write about Poland, I think it is better to put a more general background such as writing about vaccination in general or tetanus or both of them, instead of starting to write directly about the problematic in Poland.
Response 6. We thank the reviewer for this comment. A broader background on tetanus vaccination is presented later in the Introduction and in the Discussion. In this study, our primary focus is on national recommendations for tetanus vaccination and the situation in Poland. In the opening section, we aimed to explicitly emphasize this local focus from the outset. Only in the later sections of the manuscript do we address the context in other countries.
Comment 7. Line 47, please write "Coronavirus disease 19 (COVID-19)" instead of the directly the acronym, since it was the first time that this appeared in the text.
Response 7. Revised in accordance with the reviewer’s suggestion:
During the coronavirus disease 19 (COVID-19) pandemic, adult vaccination coverage against SARS-CoV-2 in Poland lagged behind Western European countries.
Comment 8. Lines 115-119, there are results, please consider moving this part to the results section or to remove it to avoid repetition.
Response 8. We agree with the reviewer’s comment. The highlighted section has been moved to the Results section.
Comment 9. Line 129, I would have appreciated the content of the 17 items of the questionnaire, maybe it would be interesting for the readers if it will be shared in Supplementary Material as an example.
Response 9. We appreciate this comment. The translated questionnaire has been included as supplementary material. We apologize for the error in the text—the questionnaire contained 16 questions, not 17. It should be noted that not all the information requested in the questionnaire was ultimately used in the study.
Comment 10. Line 140-142, please avoid copying exactly the same sentence as lines 312-314 (Institutional Review Board Statement).
Response 10. The section contained in lines 312–314 has been removed.
Comment 11. Please consider improving the quality of Figure 1 (Capitalize the first letters, avoid the overlap...etc.) . In addition, captions of figures must be put under the figure.
Response 11. Revised in accordance with the reviewer’s suggestion.
Comment 12. In the whole manuscript, please consider following the instructions to authors in the journal website to avoid minor mistakes related to the layout.
Response 12. We appreciate this comment. The manuscript has been reviewed in accordance with the guidelines.
Comment 13. Line 196, same as my comment number 5
Response 13. Revised in accordance with the reviewer’s suggestion.
Comment 14. The discussion section was well written, I would like to congratulate the authors, the structure is logical and fluid. Moreover, the limitations and perspectives are described indirectly within the flow, which will be pleasant for readers.
Response 14. We thank you for this comment.
Comment 15. Line 289, same as my comments number 5 and number 13.
Response 15. Revised in accordance with the reviewer’s suggestion.
Comment 16. Lines 305-307, please try to rephrase this part as it doesn't read well.
Response 16. We thank you for this comment. The changes have been made as outlined below.
The increasing prevalence of vaccine hesitancy in Poland—affecting both mandatory and recommended immunization programs—may have a detrimental impact on adult compliance with tetanus prophylaxis. Therefore, it is crucial to seize every clinical opportunity to educate patients and promote immunization.
Comment 17. Regarding references, if possible, please consider adding more recent references, specifically from 2025.
Response 17. We appreciate this suggestion.
Reviewer 3 Report
Comments and Suggestions for Authors
Dear Authors,
thanks for your contribuitoon.
I have comments to make regarding the study:
1. The main limitation of the study is the small sample size (N=97). This number limits the statistical power and the possibility of generalizing the results. The authors should discuss this limitation more explicitly.
2. Blood donors are a self-selected population and, as suggested by the authors themselves, may be more health conscious than the general population. This selection bias should be emphasized more strongly in the "Discussion" section, as it may explain the relatively high percentage (89.7%) of individuals with protective immunity found.
3. It would be useful to provide the 17-item questionnaire as supplementary material to increase transparency and reproducibility.
4. A brief description of recruitment arrangements could help to put the acceptance rate into context
Author Response
Comment 1. The main limitation of the study is the small sample size (N=97). This number limits the statistical power and the possibility of generalizing the results. The authors should discuss this limitation more explicitly.
Response 1. We thank the Reviewer for this valuable comment. The relatively small sample size primarily resulted from the exclusion of individuals who did not meet the study’s inclusion criteria. In addition to the age range (50–64 years), the main inclusion criterion was the absence of tetanus vaccination within the past 30 years. During the recruitment period, only 97 individuals met these criteria and consented to participate in the study. We acknowledge that a larger sample size would allow for more robust statistical analysis of the results; therefore, this study may serve as a starting point for further research.
Comment 2. Blood donors are a self-selected population and, as suggested by the authors themselves, may be more health conscious than the general population. This selection bias should be emphasized more strongly in the "Discussion" section, as it may explain the relatively high percentage (89.7%) of individuals with protective immunity found.
Response 2. We appreciate the reviewer’s thoughtful remark. We agree that blood donors represent a specific population—individuals who are aware that, by voluntarily presenting themselves at blood donation centers, they must meet numerous rigorous health requirements. Consequently, this group is indeed very well screened, including for infectious diseases. However, when designing the study, we assumed that this does not necessarily imply a high level of vaccination coverage. Assessment of post-vaccination immunity against tetanus, as well as against other diseases, is not routinely performed. Common practice involves testing only the level of anti-HBs antibodies following vaccination against hepatitis B virus. Therefore, we were interested in determining whether individuals who actively care for their health by donating blood also demonstrate a high level of awareness regarding immunization. We respectfully disagree with the reviewer’s suggestion that this constitutes a “selection bias,” as we deliberately chose this study population and took potential limitations into account.
Comment 3. It would be useful to provide the 17-item questionnaire as supplementary material to increase transparency and reproducibility.
Response 3. We appreciate this comment. The translated questionnaire has been included as supplementary material. We apologize for the error in the text—the questionnaire contained 16 questions, not 17. It should be noted that not all the information requested in the questionnaire was ultimately used in the study.
Comment 4. A brief description of recruitment arrangements could help to put the acceptance rate into context.
Response 4. We thank the reviewer for this comment. Individuals aged 50–64 years who presented at a blood donation center for the purpose of donating blood were informed about the study during the eligibility assessment and provided with an informed consent form. If, after reviewing the study’s objectives, procedures, and conditions, they gave written consent and were subsequently cleared by a physician to donate blood, a blood sample was collected to assess their post-vaccination antibody levels. All study participants whose antibody levels were deemed insufficient were notified by letter or telephone and invited to attend the Department of Paediatrics and Infectious Diseases (the study site) for a consultation with a physician, during which a booster dose of the vaccine was proposed. The description of the inclusion process and the study procedures has been expanded in the Materials and Methods section.
Reviewer 4 Report
Comments and Suggestions for Authors
This article has an unusually long title and abstract, but the text is rather limited. It merely observes that blood donation can be an opportunity to check vaccination status and possibly renew it. This is not a topic unknown to vaccination practice.
- The fact that vaccination is free in Poland is interesting, but it does not solve the problem of vaccine implementation. Tetanus is a vaccine-preventable disease that still commonly occurs in many low-income and middle-income countries, although it is rare in high-income countries. The authors correctly frame their study from the perspective of the usefulness of this vaccination for public health in Poland. They should probably explain why they only considered tetanus vaccination and didn't instead follow the co-administration strategy, which is supported by many [see: Bonanni P, Steffen R, Schelling J, Balaisyte-Jazone L, Posiuniene I, Zatoński M, Van Damme P. Vaccine co-administration in adults: An effective way to improve vaccination coverage. Hum Vaccin Immunother. 2023 Dec 31;19(1):2195786. doi: 10.1080/21645515.2023.2195786. ]. In many European countries, tetanus vaccination is administered together with the whooping cough and diphtheria vaccines. How is the procedure done in Poland?
- The study evaluated the results of the revaccination in a very small group of participants. It is important to acknowledge the limitations of this study. The article is completely missing the "Limitations" section.
- The authors reported some studies on vaccination coverage conducted on small groups of blood donors, often many years ago. They should have, more correctly, analyzed the problem of vaccination coverage, which is very sensitive in various countries, from the United States [Lu PJ, Hung MC, Srivastav A, Grohskopf LA, Kobayashi M, Harris AM, Dooling KL, Markowitz LE, Rodriguez-Lainz A, Williams WW. Surveillance of Vaccination Coverage Among Adult Populations -United States, 2018. MMWR Surveill Summ. 2021 May 14;70(3):1-26. doi: 10.15585/mmwr.ss7003a1.], the WHO [Kaur G, Danovaro-Holliday MC, Mwinnyaa G, Gacic-Dobo M, Francis L, Grevendonk J, Sodha SV, Sugerman C, Wallace A. Routine Vaccination Coverage - Worldwide, 2022. MMWR Morb Mortal Wkly Rep. 2023 Oct 27;72(43):1155-1161. doi: 10.15585/mmwr.mm7243a1.], African states [Cheng A, Ghanem-Uzqueda A, Hoff NA, Ashbaugh H, Doshi RH, Mukadi P, Budd R, Higgins SG, Randall C, Gerber S, Kabamba M, Ngoie Mwamba G, Okitolonda-Wemakoy E, Muyembe-Tanfum JJ, Rimoin AW. Tetanus seroprotection among children in the Democratic Republic of the Congo, 2013-2014. PLoS One. 2022 May 19;17(5):e0268703. doi: 10.1371/journal.pone.0268703. --- Casey RM, Nguna J, Opar B, Ampaire I, Lubwama J, Tanifum P, Zhu BP, Kisakye A, Kabwongera E, Tohme RA, Dahl BA, Ridpath AD, Scobie HM. Field investigation of high reported non-neonatal tetanus burden in Uganda, 2016-2017. Int J Epidemiol. 2023 Aug;52(4):1150-1162. doi: 10.1093/ije/dyad005. This article is an analysis of pneumococcal vaccination in the elderly in one of the 110 Italian provinces that aims to evaluate the coverage and the trend between 2018 and 2023.], Asiatic countries [Nguyen TH, Le XTT, Nguyen LH, Le HT, Do TTT, Nguyen HLT, Nguyen HT, Latkin CA, Ho CSH, Ho RCM. Resource mobilization for tetanus vaccination in Vietnam: Uptake, demand and willingness to pay among women of reproductive age. Front Public Health. 2022 Oct 18;10:980850. doi: 10.3389/fpubh.2022.980850.]. In other words, the study is not framed within the international literature.
- The authors should avoid citing studies that are more than 25 years old and report on the current situation, especially regarding the numerous attempts that have been made in various parts of the world to improve vaccination coverage.
Author Response
Comment. This article has an unusually long title and abstract, but the text is rather limited. It merely observes that blood donation can be an opportunity to check vaccination status and possibly renew it. This is not a topic unknown to vaccination practice.
The fact that vaccination is free in Poland is interesting, but it does not solve the problem of vaccine implementation. Tetanus is a vaccine-preventable disease that still commonly occurs in many low-income and middle-income countries, although it is rare in high-income countries. The authors correctly frame their study from the perspective of the usefulness of this vaccination for public health in Poland. They should probably explain why they only considered tetanus vaccination and didn't instead follow the co-administration strategy, which is supported by many [see: Bonanni P, Steffen R, Schelling J, Balaisyte-Jazone L, Posiuniene I, Zatoński M, Van Damme P. Vaccine co-administration in adults: An effective way to improve vaccination coverage. Hum Vaccin Immunother. 2023 Dec 31;19(1):2195786. doi: 10.1080/21645515.2023.2195786. ]. In many European countries, tetanus vaccination is administered together with the whooping cough and diphtheria vaccines. How is the procedure done in Poland?
The study evaluated the results of the revaccination in a very small group of participants. It is important to acknowledge the limitations of this study. The article is completely missing the "Limitations" section.
The authors reported some studies on vaccination coverage conducted on small groups of blood donors, often many years ago. They should have, more correctly, analyzed the problem of vaccination coverage, which is very sensitive in various countries, from the United States [Lu PJ, Hung MC, Srivastav A, Grohskopf LA, Kobayashi M, Harris AM, Dooling KL, Markowitz LE, Rodriguez-Lainz A, Williams WW. Surveillance of Vaccination Coverage Among Adult Populations -United States, 2018. MMWR Surveill Summ. 2021 May 14;70(3):1-26. doi: 10.15585/mmwr.ss7003a1.], the WHO [Kaur G, Danovaro-Holliday MC, Mwinnyaa G, Gacic-Dobo M, Francis L, Grevendonk J, Sodha SV, Sugerman C, Wallace A. Routine Vaccination Coverage - Worldwide, 2022. MMWR Morb Mortal Wkly Rep. 2023 Oct 27;72(43):1155-1161. doi: 10.15585/mmwr.mm7243a1.], African states [Cheng A, Ghanem-Uzqueda A, Hoff NA, Ashbaugh H, Doshi RH, Mukadi P, Budd R, Higgins SG, Randall C, Gerber S, Kabamba M, Ngoie Mwamba G, Okitolonda-Wemakoy E, Muyembe-Tanfum JJ, Rimoin AW. Tetanus seroprotection among children in the Democratic Republic of the Congo, 2013-2014. PLoS One. 2022 May 19;17(5):e0268703. doi: 10.1371/journal.pone.0268703. --- Casey RM, Nguna J, Opar B, Ampaire I, Lubwama J, Tanifum P, Zhu BP, Kisakye A, Kabwongera E, Tohme RA, Dahl BA, Ridpath AD, Scobie HM. Field investigation of high reported non-neonatal tetanus burden in Uganda, 2016-2017. Int J Epidemiol. 2023 Aug;52(4):1150-1162. doi: 10.1093/ije/dyad005. This article is an analysis of pneumococcal vaccination in the elderly in one of the 110 Italian provinces that aims to evaluate the coverage and the trend between 2018 and 2023.], Asiatic countries [Nguyen TH, Le XTT, Nguyen LH, Le HT, Do TTT, Nguyen HLT, Nguyen HT, Latkin CA, Ho CSH, Ho RCM. Resource mobilization for tetanus vaccination in Vietnam: Uptake, demand and willingness to pay among women of reproductive age. Front Public Health. 2022 Oct 18;10:980850. doi: 10.3389/fpubh.2022.980850.]. In other words, the study is not framed within the international literature.
The authors should avoid citing studies that are more than 25 years old and report on the current situation, especially regarding the numerous attempts that have been made in various parts of the world to improve vaccination coverage.
Response. We thank the reviewer for the numerous comments. A Limitations of the study section has been added, as presented below.
Our study has certain limitations and constraints. First and foremost, the study population was relatively small, consisting of only 97 participants. This limited sample size was primarily due to the strict inclusion criteria. In addition to the age requirement (50–64 years), a key criterion was the absence of tetanus vaccination within the past 30 years. During the recruitment period, 97 individuals who met these criteria and provided informed consent were enrolled. We acknowledge that a larger sample size would have enhanced the statistical power of our analysis and allowed for broader generalization of the findings. Nevertheless, we consider this study a potential starting point for further research.
Another limitation may stem from the specific nature of the study population. Blood donors represent a self-selected group who are typically aware of the strict health requirements necessary for donation eligibility. As such, this population is routinely screened for various health conditions, including infectious diseases. However, when designing the study, we assumed that this heightened health awareness does not necessarily correlate with high vaccination coverage. Post-vaccination immunity against tetanus – as well as against other vaccine-preventable diseases – is not routinely assessed. In clinical practice, only anti-HBs antibody levels are commonly measured following hepatitis B vaccination. Therefore, we were interested in determining whether individuals who demonstrate proactive health behaviors through regular blood donation also exhibit a strong commitment to preventive care.
Despite the limitations mentioned above, we believe that our study provides valuable data on tetanus post-vaccination immunity among adults in Poland. To date, no prospective study has assessed whether – and how – the blood donation process could serve as an opportunity to evaluate and potentially improve post-vaccination immunity.
We thank the reviewer for the detailed feedback. While this study may not directly align with the existing international literature, this is precisely because it is original in its design. Although recommendations for using the opportunity of blood donation to administer vaccinations may exist, to our knowledge, this concept has not been prospectively evaluated. We also considered voluntary blood donors to be a group with greater awareness of the importance of preventive healthcare.
The decision to focus our study on tetanus vaccination was based on the following considerations:
In Poland, tetanus vaccination is mandatory and provided free of charge. The combined Tdap vaccine is recommended but fully self-funded, which could negatively influence willingness to receive a booster dose. In our study, we aimed to explore ways to improve vaccination coverage within the existing preventive healthcare system. For adults without additional risk factors, vaccinations other than Tdap are rarely recommended before the age of 60. In Poland, first-time blood donors are accepted up to the age of 60, while repeat donors may donate until the age of 65.
2a. Measuring anti-tetanus toxoid antibody concentrations was intended as an additional motivating factor for vaccination. All individuals who agreed to participate were interested in knowing their immune status and were clearly informed about it. Despite receiving alarming results, being counseled by a physician, being qualified for vaccination (47.5% of participants), and being offered a convenient opportunity to receive the vaccine, only 17.5% took advantage of it. To our knowledge, this has not previously been investigated.
2b. Assessment of tetanus immunity, even after many years, is straightforward and highly reliable from a clinical standpoint. By contrast, antibody levels against pertussis toxin decline rapidly after vaccination and are not suitable for assessing immune status—the date of the last booster dose remains the only reliable indicator.
Round 2
Reviewer 1 Report
Comments and Suggestions for Authors
Accept in present form
Reviewer 3 Report
Comments and Suggestions for Authors
Dear Authors,
Thank you for responding point by point to the suggestions
The manuscript thus reworked is now clearer and refined
Reviewer 4 Report
Comments and Suggestions for Authors
The reviewers revised the manuscript